Predicting hedgehog mortality risks on British roads using habitat suitability modelling

Wright Patrick G.R. 1 2
Coomber Frazer G. 1 2
http://orcid.org/0000-0002-3830-0995 Bellamy Chloe C. 3
http://orcid.org/0000-0002-7457-2699 Perkins Sarah E. 4
http://orcid.org/0000-0002-2580-2769 Mathews Fiona 1 2 f.mathews@sussex.ac.uk
1 Life Sciences, University of Sussex , Brighton , UK
2 The Mammal Society , London , UK
3 Centre for Ecosystems, Forest Research , Roslin , UK
4 School of Biosciences, Cardiff University , Cardiff , UK
Hedrick Ann
Electronic publication date: 2020 Jan 21
Publication date: 2020
Volume: 7
Electronic Location ID: e8154
Received 2019 Jul 12; Accepted 2019 Nov 4
Copyright: © 2019 Wright et al.
Copyright year: 2019
Copyright holder: Wright et al.
License: This is an open access article distributed under the terms of the Creative Commons Attribution License, which permits unrestricted use, distribution, reproduction and adaptation in any medium and for any purpose provided that it is properly attributed. For attribution, the original author(s), title, publication source (PeerJ) and either DOI or URL of the article must be cited.
License URL: https://creativecommons.org/licenses/by/4.0/

Keywords: Erinaceus europaeus, Wildlife-vehicle collisions, Population decline, Mitigation, Roadkill, Maxent, Habitat fragmentation, Hedgehogs, Road casualties

Funding: Mammal Society, the People’s Trust for Endangered Species and The British Hedgehog Preservation Society NERC KE Fellowship NE/S006486/1 University of Sussex This work was supported by the Mammal Society, the People’s Trust for Endangered Species and The British Hedgehog Preservation Society. Fiona Mathews is supported by a NERC KE Fellowship NE/S006486/1 and the University of Sussex. Frazer Coomber is supported by the University of Sussex. The initial study design was developed by the authors and the funders who were interested in using the roadkill presence data available to better understand where they are occurring. The authors were free to write up the results without interference. The decision to publish the manuscript also lay with the authors.

==============================
Road vehicle collisions are likely to be an important contributory factor in the decline of the European hedgehog (Erinaceus europaeus) in Britain. Here, a collaborative roadkill dataset collected from multiple projects across Britain was used to assess when, where and why hedgehog roadkill are more likely to occur. Seasonal trends were assessed using a Generalized Additive Model. There were few casualties in winter—the hibernation season for hedgehogs—with a gradual increase from February that reached a peak in July before declining thereafter. A sequential multi-level Habitat Suitability Modelling (HSM) framework was then used to identify areas showing a high probability of hedgehog roadkill occurrence throughout the entire British road network (∼400,000 km) based on multi-scale environmental determinants. The HSM predicted that grassland and urban habitat coverage were important in predicting the probability of roadkill at a national scale. Probabilities peaked at approximately 50% urban cover at a one km scale and increased linearly with grassland cover (improved and rough grassland). Areas predicted to experience high probabilities of hedgehog roadkill occurrence were therefore in urban and suburban environments, that is, where a mix of urban and grassland habitats occur. These areas covered 9% of the total British road network. In combination with information on the frequency with which particular locations have hedgehog road casualties, the framework can help to identify priority areas for mitigation measures.

Introduction

Growing human populations, and associated urbanisation, have led to an expansion of road networks and traffic volumes globally. In Europe, an average of 70,000 km of roads are built every year, while in Britain, traffic on major roads has increased by 9.5% since 2000 and is expected to grow an estimated 17–51% by 2050 (UK-Gov, 2018).

Road traffic has both direct and indirect impacts on wildlife (Van Der Ree et al., 2011; Rytwinski & Fahrig, 2015; Bennett, 2017). These include the loss and fragmentation of habitat (Andrews, 1990; Eigenbrod, Hecnar & Fahrig, 2008; Ibisch et al., 2016), disturbance via elevated noise and pollution levels (Berthinussen & Altringham, 2012) and direct mortality through wildlife-vehicle collisions (WVCs), which can, in turn, drive population declines (Fahrig & Rytwinski, 2009). Several studies have been undertaken to understand the threats posed by roads to wildlife and to identify potential mitigation measures, although most published evidence is derived from regional case studies (Van Der Ree et al., 2011).

WVCs are known to occur for a wide range of mammal species (Barthelmess & Brooks, 2010). Numerous studies have shown the negative impact of roadkill on populations of large carnivores, such as tigers (Panthera tigris altaica; Kerley et al., 2002), grizzly bears (Ursus arctos; Waller & Servheen, 2005) and servals (Leptailurus serval; Williams et al., 2019). Collisions likely to pose a risk to human safety (e.g. with large mammals such as deer) are also a focus of research (Girardet, Conruyt-Rogeon & Foltête, 2015). The frequency of these collisions appears to be increasing, possibly owing to the global increases in traffic volume (Madsen, Strandgaard & Prang, 2002). Small mammal species (<1 kg in body mass), however, often go under recorded in roadkill surveys due to their lower detectability (Slater, 2002; Ford & Fahrig, 2007; Langen et al., 2007) and the fact that their carcases are rapidly scavenged (Slater, 2002; Barthelmess & Brooks, 2010; Schwartz et al., 2018). This hinders the ability to identify factors that are associated with roadkill, and to develop targeted and effective mitigation measures that aim to limit the impact of roads on wildlife populations through either a modification of motorist behaviour and/or the modification of animal behaviour (Glista, DeVault & DeWoody, 2009). These measures are often species- or genus-specific, and vary in their forms (e.g. wildlife warning signs, wildlife crossings, fencing, under- & overpasses) and effectiveness in different situations (Clevenger & Waltho, 2000; Glista, DeVault & DeWoody, 2009; Rytwinski et al., 2016).

The likelihood of WVCs is often linked to local road characteristics and the surrounding landscape (Neumann et al., 2012; Grilo et al., 2016; Santos et al., 2018). Habitat Suitability Models (HSMs) can predict and examine the relationship between the distribution of animal observations and environmental variables through mathematical algorithms (Guisan & Zimmermann, 2000; Elith & Leathwick, 2009). These models can be used to identify high quality habitat and derive information on the underlying environmental drivers (Phillips, Anderson & Schapire, 2006; Elith et al., 2011). Although HSMs are typically used to identify and predict species’ habitat requirements (Razgour, Hanmer & Jones, 2011; Croft, Chauvenet & Smith, 2017), they have also been used previously to identify the environmental variables that explain and predict roadkill occurrence (Malo, Suarez & Diez, 2004; Santos et al., 2013; Fabrizio et al., 2019). For example Ha & Shilling (2018) successfully characterised the relationship between forest cover and road density on ungulates roadkill in central California.

To create HSMs and other models to better understand WVC patterns, some prior, spatially explicit species information is required. Biological records—reports of species or taxa at a given location and time—can be collected via bespoke surveys. However, this approach is resource intensive and expensive. Biological records from citizen science studies, where volunteers collect data as part of a scientific study and as ad hoc observations, represent a valuable source of ecological data (Powney & Isaac, 2015). Recent technological advancements, such as online reporting and smart phone apps, have facilitated a huge growth in the adoption of this approach in various fields (Silvertown, 2009; Pocock et al., 2015). Numerous citizen science approaches have already been established for recording and surveying roadkill (Heigl & Zaller, 2014; Vercayie & Herremans, 2015; Heigl et al., 2016); the resulting data have been shown to provide useful spatial information on roadkill clusters, despite varying slightly from data collected via comparable surveys run by trained observers (Périquet et al., 2018). However, biological records from citizen science projects or ad hoc observations are typically unstructured and prone to biases and errors owing to issues such as a lack of standardised survey protocols (Prendergast et al., 1993; Isaac et al., 2014). Noisy data can hide, or increase uncertainty around, true trends, and biases can create false relationships (Kamp et al., 2016). However, if records are carefully filtered to remove those with geographic uncertainty, and if sampling bias is effectively accounted for during analysis, biological records represent a huge data resource (Kery et al., 2010; Hill, 2012; Van Strien, Van Swaay & Termaat, 2013; Isaac et al., 2014).

The aim of this study is to investigate the spatial and temporal distribution of European hedgehog (Erinaceus europaeus) roadkill records across the entire British road network. The European hedgehog is widespread throughout most of Europe, but is thought to be in decline in many countries. In Britain, for example population estimates have reduced from 1.5 million individuals in 1995 to 522,000 in 2016 (Mathews et al., 2018). The causes for the species’ dramatic decline remain unclear, but factors including the intensification of agriculture resulting in a loss of habitat and prey (Hof & Bright, 2009), predation and competition with badgers (Pettett et al., 2018; Williams et al., 2018) and road collisions (Huijser & Bergers, 2000; Rondinini & Doncaster, 2002) are likely to contribute. It has been estimated that approximately 100,000 to 300,000 hedgehogs are killed every year on roads in Britain (Roos, Johnston & Noble, 2012; Wembridge et al., 2016), Belgium (Holsbeek, Rodts & Muyldermans, 1999) and the Netherlands (Huijser, Bergers & De Vries, 1998). Yet, the causal factors behind these large numbers of roadkill are still poorly understood.

Here, we analysed seasonal trends in hedgehog road casualties and, by using a multi-level HSM framework, produced spatial predictions of the probability of hedgehog road casualties across Britain, based on environmental variables. Furthermore, the analysis of temporal trends was used to provide seasonal patterns of hedgehog roadkill risk and to highlight months when the majority of roadkill are recorded. The outputs can inform targeted mitigation measures aiming at reducing the impact of roads on the declining hedgehog population in Britain.

Materials and Methods

Hedgehog roadkill data

Data on hedgehog roadkill were obtained from multiple citizen science projects in Britain (Big Hedgehog Map, Mammal Mapper, Mammal Tracker, Mammals on Roads, Project Splatter and Suffolk Wildlife Trust), National Biodiversity Network gateway and national and local monitoring schemes (Supplemental Information 1). These records were collections of both transect associated and opportunistic records, collected using a variety of recording methods, including online recording forms, smart phone apps and email submitted records. A total of 12,684 hedgehog roadkill records were collected from 1959 to 2018.

Each record was subject to screening against inclusion criteria before being considered in the models. Only confirmed roadkill records were used, for example, live hedgehogs from projects that did not specifically target roads were excluded. Similarly, information on recorder effort (e.g. transects) was not necessary and therefore not retained. Due to a very low number of records before 2000, only records made between January 2000 and December 2017 were included, aligning with the environmental data used. For each HSM level (1 km or 100 m), only records with a spatial precision equal to or finer than the model resolution were retained, and only single record per grid cell was used. Similarly, the seasonality model only included records that had dates with information on at least month and year.

Modelling hedgehog roadkill seasonality

A Generalized Additive Model (GAM) was used to characterise the seasonal trends in hedgehog roadkill throughout the year using the ‘mgcv’ package (Wood & Wood, 2015) in the statistical analysis software R (v.3.4.3; R Core Team, 2018), implemented in R Studio (v.1.1.456; RStudio Team, 2018). The monthly number of hedgehog roadkill was modelled using a Poisson family error distribution. Five candidate models were fitted using a cyclic cubic spline, with 12 knots for the month variable. The best model was selected based on a generalised likelihood ratio test. The autocorrelation function and partial autocorrelation function were used to assess the residuals of all models. The first candidate model assumed that all observations were independent. The next four candidate models included a correlation argument for the month variable, within an extended mixed model framework of the GAM, ranging from 1 to 4, to account for temporal non-independence.

Modelling probability of hedgehog roadkill occurrence

The probability of hedgehog roadkill occurrence on British roads was modelled using the presence-only modelling software MaxEnt (Phillips, Anderson & Schapire, 2006). MaxEnt determines occurrence probabilities by using a set of explanatory variables together with species’ presence data and expresses the suitability of each grid cell as a function of the explanatory variables within that grid cell (Phillips, Anderson & Schapire, 2006). The filtered hedgehog roadkill data were used as the presence data within all subsequent models (Fig. 1). A suite of candidate environmental explanatory variables was selected based on a priori expectations from the literature that they influence the presence or behaviour of hedgehogs (Driezen et al., 2007; Croft, Chauvenet & Smith, 2017). The source and resolution of the candidate explanatory variables is provided in Table 1 and further information on their preparation can be found in Supplemental Information 2.

Figure 1 Location of the hedgehog roadkill presence data used for the sequential multi-level Habitat Suitability Modelling (HSM).

(A) The national scale model. (B) Major roads. (C) B road. (D) Minor roads. (E) All roads.

Table 1 Summary of the explanatory variables used for the HSMs.

The national models were performed at a 1–3 km resolution, while the road models were performed at a 100–500 m resolution.

Explanatory variable	Resolution	Source	
Arable (% cover)	1–3 km	Land cover 2007 (LCM: Morton et al., 2011)	
Broadleaved (% cover)	1–3 km	Land cover 2007 (LCM: Morton et al., 2011)	
Coniferous woodland (% cover)	1–3 km	Land cover 2007 (LCM: Morton et al., 2011)	
Urban (% cover)	1–3 km	Land cover 2007 (LCM: Morton et al., 2011)	
Improved grassland (% cover)	1–3 km	Land cover 2007 (LCM: Morton et al., 2011)	
Rough grassland (% cover)	1–3 km	Land cover 2007 (LCM: Morton et al., 2011)	
Freshwater (% cover)	1–3 km	Land cover 2007 (LCM: Morton et al., 2011)	
Greenspace (% cover)	1–3 km	OS MasterMap Topography Layer (2019)	
Hedgerow density	1–3 km	Woody linear features (Scholefield et al., 2016)	
Slope	1–3 km	SRTM slope DEM (Pope, 2017)	
All road traffic	1–3 km	data.gov.uk (2019)	
B road density	1–3 km	OS Open Roads (2019)	
Major road density	1–3 km	OS Open Roads (2019)	
Major road traffic	1–3 km	data.gov.uk (2019)	
Minor road density	1–3 km	OS Open Roads (2019)	
Minor road traffic	1–3 km	data.gov.uk (2019)	
Population density	1–3 km	EEA (Gallego, 2010)	
Summer precipitation	1–3 km	Worldclim—global climate data (Fick & Hijmans, 2017)	
Summer temperature	1–3 km	Worldclim—global climate data (Fick & Hijmans, 2017)	
Distance from arable (m)	100–500 m	Land cover 2007 (LCM: Morton et al., 2011)	
Distance from broadleaved woodland (m)	100–500 m	Land cover 2007 (LCM: Morton et al., 2011)	
Distance from Greenspace (m)	100–500 m	Land cover 2007 (LCM: Morton et al., 2011)	
Distance from buildings (m)	100–500 m	Land cover 2007 (LCM: Morton et al., 2011)	
Distance from improved grassland (m)	100–500 m	Land cover 2007 (LCM: Morton et al., 2011)	
Distance from urban area (m)	100–500 m	Land cover 2007 (LCM: Morton et al., 2011)	
Distance from suburban area (m)	100–500 m	Land cover 2007 (LCM: Morton et al., 2011)	
Distance from major road*	100–500 m	OS Open Roads (2019)	
Distance from B road*	100–500 m	OS Open Roads (2019)	
Distance from minor road*	100–500 m	OS Open Roads (2019)	
Slope	100–500 m	SRTM slope DEM (Pope, 2017)	
HSM (1–3 km)	100–500 m	–	

The MaxEnt models were created and integrated across two levels using a multi-level HSM framework developed by Bellamy (in press). This framework models species’ distributions at multiple geographic levels in turn, from coarsest to finest resolutions, whilst including habitat suitability indices (HSI) from the preceding coarser resolution level as an explanatory variable in models at a subsequent model level. This sequential approach reduces predictor collinearity and enables a species’ response to the local environment at lower levels to vary according to higher level conditions (Bellamy, in press). The modelling process for hedgehog roadkill was set out as below: A national-level model for the whole of Britain was constructed at a one km grid cell resolution. It focused on percentage cover of habitat variables, road density and climatic variables, which were measured at candidate resolutions of 1–3 km (Table 1; Supplemental Information 2).

Road-level models were constructed at a higher, 100 m grid cell resolution and covered all areas in Britain within 200 m of the road. Road types considered were based on the official and unofficial road type classifications of the Department of Transport (Department for Transport, 2002; refer to Supplemental Information 3 for full description and classifications of road types). Each individual road-level model was constructed according to the following road types: Minor roads (minor roads and local roads)

B roads

Major roads (A roads and motorways)

All roads combined.

The explanatory variables associated with the individual road-level models were primarily associated with distance to landscape features (e.g. distance from broadleaved woodland). All explanatory variables were measured at two candidate scales: 100 m and 500 m. Each variable was cropped within a 200 m buffer for each road type.

The national-level model was run first and the HSI values for the probability of hedgehog roadkill occurrence were calculated. These values were then incorporated as an explanatory variable into the subsequent higher resolution road-level models, as described above, by disaggregating them to the resolution of road-level models. The optimal resolution for each variable was identified at each level by comparing the univariate MaxEnt model (default settings). The resolution which resulted in the model with the highest measure of training gain (interpreted as the likelihood of the presence points; Merow, Smith & Silander, 2013), was retained for the final model. For each level, highly correlated variables were removed using the ‘vifstep’ stepwise function in the ‘usdm’ package (Naimi et al., 2014) based on a conservative VIF threshold of three (Zuur, Ieno & Elphick, 2010). The package ‘ENMeval’ (Muscarella et al., 2014) was used to identify the optimal MaxEnt model settings. Combinations of feature types were tested with the threshold features disabled to reduce overfitting (L, linear; H, hinge; Q; quadratic; P, product): L, H, LQ, LQH and LQHP. The regularisation multiplier was varied in steps of 0.5, from 0.5 to 4. These optimal settings were then used to produce model predictions. All analyses were carried out using R (v.3.4.3; R Core Team, 2018) in R Studio (v.1.1.456; RStudio Team, 2018).

The HSIs from each HSM were used to classify grid cells with the highest probability of hedgehog roadkill occurrence. The predicted HSI values were partitioned into a binary response (high and low probability of hedgehog roadkill occurrence) using the Maximum Training Sensitivity and Specificity (MTSS) occupancy rule (Liu, White & Newell, 2013).

In order to compare hedgehog roadkill probabilities between the different road types, the mean HSI value of major road, B road and minor road-level models were extracted directly from the ‘all roads’ model. To take into account differences in the availability of different road types, a ratio of the probability of occurrence relative to the length of each road type was computed. The ratio was calculated as the number of 100 m2 grid cells classed as having a high probability of roadkill occurrence divided by the number of grid cells classified as having a low probability of roadkill occurrence (based on binary model outputs).

To control for possible bias in the hedgehog roadkill records towards locations that were heavily observed, pseudo-absence points (background points) were used for the HSMs based on the combined roadkill records of seven other mammal species. The species used (Vulpes vulpes, Neovison vison, Mustela erminea, Mustela nivalis, Meles meles, Lutra lutra and Capreolus capreolus) all shared a similar national distribution and are frequently recorded as roadkill. These records were from the same time period and underwent the same inclusion criteria as the hedgehog dataset. For the road-level models, subsets were created by including only records that were within the 200 m road buffers. HSMs were also run using 10,000 (national model) and 2,500 (road models) random pseudo-absence points. The degrees of similarity between the roadkill and random pseudo-absence HSMs were assessed using Schoener’s index. HSM performance was judged using the Area Under Curve (AUC) test statistic, with one being perfect predictions and 0.5 representing a model that does not predict better than random chance.

To test for variations between the observed hedgehog roadkill records and the locations predicted to have a high probability of occurrence, three maps were also produced: The inter-annual persistence of observed roadkill was calculated based on the number of years where at least one hedgehog roadkill per year was recorded in each 10 km British National Grid cell.

Kernel density at a 10 km resolution of the number of roadkill presence records used for the national-level model.

Kernel density at a 10 km resolution of grid cells classified as having a high probability of hedgehog roadkill occurrence based on the ‘all roads’ HSM binary results.

The relationship between all three outputs was then assessed using pairwise Pearson’s correlation.

Results

After the inclusion criteria were applied to all the hedgehog roadkill data, a total of 6,004 records were available for all subsequent HSMs (Fig. 1) and 3,777 for the seasonal analysis. Records were spread throughout Britain and displayed distinct areas of high and low recording (Fig. 1).

The seasonal analysis used the GAM model with a correlation argument of two, this being the best candidate model based on the Generalised Likelihood Ratio Test. The inclusion of a correlation argument took into account the temporal correlations between the monthly roadkill records. The GAM indicated clear seasonal variations in the relationship between the months of the year and the number of hedgehog roadkill recorded (F9, 24= 124.98, p < 0.001). The smoothing function indicated that roadkill was at its lowest in winter, gradually increased to a peak in July, then subsequently decreased (Fig. 2).

Figure 2 Seasonal variations in the number of hedgehog roadkill using a smoothing function (S) obtained by a generalised additive model (GAM).

All models were adequate, based on the AUC scores, but some performed better than others did. The national, minor and all road models performed well with AUC scores higher than 0.70, while the major road and B road models performed less well (AUC ≤ 0.6). High niche overlap between models using roadkill and random pseudo-absence background data was identified with all Schoener’s D statistics being >0.85 (Table 2).

Table 2 Model performance of the final national scale (1–3 km) and the road (100–500 m) roadkill HSMs using roadkill background data.

Schoener’s D index scores represent the niche overlap between models using random and roadkill background data.

Model type	N	Feature types	Regularisation multiplier	AICc	Full AUC	Schoener’s D	
National (1 km)	6,004	LQ	1	138,246	0.72	0.85	
Major road (100 m)	788	LQ	1	22,054	0.5	0.9	
B road (100 m)	368	LQH	4	10,186	0.6	0.84	
Minor road (100 m)	1,476	LQH	4	44,442	0.71	0.93	
All roads (100 m)	2,623	LQHPT	4	79,724	0.74	0.92	

The shape and strength of the species-environment relationships varied between HSMs. The output response curves and permutation importance values varied between the different model levels and the different pseudo-absence data used, roadkill or random (Fig. 3; Supplemental Informations 4–18). However, certain variables, such as those associated with urban cover, buildings and grasslands, had a consistent importance throughout all models.

Figure 3 Mapped logistic habitat suitability indices (HSI) for E. europaeus roadkill at a national-level and road-level using all roads.

Mapped logistic HSI for E. europaeus roadkill at a national-level (A), and road-level using all road data (B) according to the hierarchical, multi-scale model using roadkill background data. The response curves with the highest permutation importance are represented for the national-level (C), (D), (E) and (F) and the model using all road data (G), (H), (I) and (J).

The probability of hedgehog roadkill occurrence in the national-level model using roadkill pseudo-absences was characterised by medium to high urban cover (permutation importance: 68% at a one km scale), with probability of occurrence reaching a peak at approximately 50% urban cover (Fig. 3). Grassland cover was also important (permutation importance: ∼10% at a one km scale for improved grassland and a three km scale for rough grassland) with the probability of roadkill occurrence increasing with the amount of improved and rough grassland coverage (Figs. 3A and 3C–3F). Similar results were also found with the national-level model that used random pseudo-absences (Supplemental Informations 4 and 7).

The road-level model with roadkill pseudo-absences identified distance from major roads, B roads and buildings at a 100 m scale as important predictors, alongside distance from rough grassland at a 500 m scale (Figs. 3B and 3G–3J). While the outputs of the different road-level models showed high overlap (Table 2), the permutation importance of the national-level HSI varied considerably according to which background dataset was used (roadkill or random). The HSI variables from the national-level model had a high importance (∼50%) when using random points and very low importance (<5%) when using the pseudo-absence roadkill data (Fig. 3; Supplemental Information 5).

The mean HSI value extracted from the road-level model containing all roads (100 m resolution) indicated a higher probability of hedgehog roadkill occurrence on major roads (0.48) compared to minor roads (0.35) and B roads (0.28). The same pattern was observed in analyses that accounted for the differing availability of the various road types: major roads had a higher probability of roadkill occurrence ratio (0.52) than that of B roads (0.38) and minor roads (0.11).

The binary response (high and low probability of a roadkill occurence) using the MTSS identified a total of 9% of the 100 m2 grid cells from the road network, based on the ‘all roads’ model, as having a high probability of hedgehog roadkill occurrence. These areas were mainly in central Britain, southern Wales, the outskirts of London, and southern Scotland (Fig. 4). The outputs also regularly identified large areas of small towns, while only the edges of large cities had a high probability of roadkill occurrence (Fig. 4).

Figure 4 Binary output of logistic habitat suitability indices (HSI) for E. europaeus roadkill using all road data.

Red indicates areas with high probability of hedgehog roadkill occurrence and grey lines indicate roads (all types).

High levels of correlation were identified between the persistence of records over time and the kernel density of roadkill records (10 km resolution) (r = 0.93; Fig. 5). However, the kernel density of 100 m resolution grid cells predicted to have a high probability of roadkill occurrence showed a lower correlation with both the persistence and presence data (r ≤ 0.55; Fig. 5). The HSM identified regions of high probability of roadkill that were not identified from the presence records alone, such as the Central Belt (southern Scotland). It also helped to illustrate that areas with a high number of records do not necessarily identify areas with a high probability of hedgehog roadkill occurrence on a national scale, for example eastern England (Fig. 5).

Figure 5 Map representing the correlation between the roadkill presence data and HSM model predictions.

(A) Persistence of records in 10 km grid cells since 2000. (B) Kernel density of roadkill records since 2000. (C) Density of cells predicted as having a high probability of hedgehog roadkill occurrence (binary outputs) from the all road model. The Pearson correlation between each map is indicated by the arrows.

Discussion

Vehicle collisions are a major threat contributing to the decline of hedgehogs in Britain. In this study, roadkill data were analysed using GAMs and multi-level HSMs to understand seasonal trends, assess the influence of landscape variables and to identify areas with a high probability of roadkill occurrence. These analyses were conducted throughout the entire British road network (∼400,000 km) at a national scale (1 km resolution) but also at a finer resolution (100 m) within 200 m of roads.

The seasonal trends in hedgehog roadkill, with low probability in winter and higher probability in summer, aligns with expectations from hedgehog biology and behaviour: hedgehogs in Britain typically hibernate between October/November to March/April (Reeve, 1994), whereas they are active largely between March and October in England (Reeve, 1981). The single peak of roadkill in July is similar to the findings of Canova & Balestrieri (2019) and Orlowski & Nowak (2006), who found peak distributions of hedgehog roadkill in July and August in Northern Italy and Poland respectively, and suggests that roadkill is likely to be linked with breeding, rather than night length. The results could indicate that most British hedgehogs have a single litter, as roadkill peaks are expected to coincide with population peaks and with the dispersal of vulnerable juvenile hedgehogs from their natal area. Mitigation measures are therefore likely to be most successful if they are implemented during these months when hedgehogs are more vulnerable.

Hedgehog roadkill probability is linked to a combination of environmental and abiotic variables. The following habitat variables were important predictors of hedgehog roadkill: the extent of improved grassland; the extent of rough grassland; urban land use; and road density (combination of distance from major and B roads). This aligns with studies of other wildlife that suggest that collision probability increases where there is a combination of favourable habitat and human dominated areas (Neumann et al., 2012; Santos et al., 2018).

This study’s finding of increased probability of hedgehog roadkill occurrence on major roads corresponds with earlier research (Bright, George & Balmforth, 2005). The results, however, cannot distinguish whether the probability of hedgehog casualties is determined by habitat per se, by high hedgehog densities associated with these habitats, or a combination of the two. The recent population review of British mammals (Mathews et al., 2018) found that unimproved (rough) grassland and arable habitats made substantial contributions to the total population estimate, with over 55% of the total estimated British hedgehog population being associated with these two habitats. Similarly, Croft, Chauvenet & Smith (2017) predicted that hedgehog abundance was highest in improved grassland and arable habitats, supporting this study’s findings that arable, improved, and rough grassland, were listed in the top four influential habitats (Supplemental Information 9). Croft, Chauvenet & Smith (2017) also reported very high HSI for urban habitats, followed equally by suburban and broadleaved woodlands. Almost 40% of the estimated population of hedgehogs in Britain was derived from broadleaved woodland habitat (Mathews et al., 2018). However, this habitat is not a notably influential factor in the current hedgehog roadkill models. Recorder efficiency may be reduced in roads that pass through woodlands as fallen leaves, shade, dappled light, and increased scavenging rates may reduce the visibility of hedgehog roadkill. Alternatively the findings may suggest that the densities of hedgehogs in deciduous woodland are much lower than expected.

The models have predicted areas of the British road network that are likely to have high probability of hedgehog roadkill. Both the national and road-level models identified central Britain as being a large area where hedgehog collisions are highly probable. This is the region where there is the most persistence (i.e. inter-annual records) of roadkill records since 2000. Urbanisation levels are also known to be a significant barrier for the range expansion of both polecats (Mustela putorius) and pine martens (Martes martes) in this region (Vincent Wildlife Trust, 2019). At a local scale, the prediction maps suggest that elevated roadkill probability occurs throughout small villages and towns, but is restricted to the suburban areas of larger cities, owing to the relative lack of grassland in the central parts of large cities. Similarly, Orlowski & Nowak (2006), in a small scale and largely rural study, found that most roadkill occurred inside villages.

The identification of regions with elevated probability of hedgehog roadkill occurrence provides the opportunity to target mitigation and to test its effectiveness. The finding that only 9% of the whole road network is identified as ‘high risk’ for hedgehogs provides an initial focus for interventions and we recommend wider-scale collection of data at high spatial resolution to improve the models further. For example at a national scale, mitigation may be best targeted on central Britain, as this includes many areas of high probability of occurrence and with repeated records across multiple years. At a local scale, high resolution models of roads can help identify those roads—possibly even locations within roads—where mitigation measures may be the most effective (See Supplemental Information 19 for Chesterfield case study). It is notable that in this example nearly all areas that have multiple reports of hedgehog roadkills across time coincide with the areas predicted to have a high probability of roadkill.

Mitigation measures have been shown to be a cost effective tool in reducing the number and effect of roadkills, for example with large ungulates in the United States and Canada (Huijser et al., 2009). Many measures will work best when engaging with communities and influencing national policies, through increased awareness and effectively implemented management plans (Dickman, 2010). There are a variety of mitigation options that could be used to lower the probability of hedgehog roadkill (Glista, DeVault & DeWoody, 2009; Rytwinski et al., 2016). Hedgehog roadkill probability was positively associated with good habitat, it is important to ensure that mitigation measures ensure safe crossing of roads rather than dissuading hedgehogs to use such habitats. Road signs and speed reductions can be used to alert drivers when approaching areas of high risk. Although this type of intervention does not affect all species equally (Dique et al., 2003), it was found to decrease overall roadkill (including echidna, a species with similar ecological traits to the European hedgehog) in Tasmania by ∼50% through a 20% speed reduction (Hobday & Minstrell, 2008). Fencing could help channel hedgehog movement away from roads, although other structures that improve connectivity by providing safe corridors across roads, such as overbridges, underpasses and culverts, may be preferred (Van Vuurde & Van Der Grift, 2005). It must be emphasised that the effectiveness of many of these structural mitigation measures still remains unclear (Rytwinski et al., 2016) and are largely determined by a few successful examples (Glista, DeVault & DeWoody, 2009). The continued recording of roadkill at high spatial resolution, before and after the implementation of mitigation measures, can provide important evidence on the effectiveness of the measures taken.

It is vital that future monitoring is conducted at a high spatial resolution: a large number of data points supplied to this project had to be discarded prior to analysis based on precision filtering (>50% of the records used for the national model were discarded for the ‘all roads’ model and >85% of records were discarded for the major and B road models). This may in turn, result in the low performance of models and low AUC values, as observed with the B road and major road models. Moreover, citizen science records should be supplemented by data from dedicated surveys, so that true absence data, rather than pseudo-absence, can be used to construct models (Brotons et al., 2004; Fiedler et al., 2018). The HSMs could also be refined in the future, when datasets are sufficiently large, by incorporating potentially important variables such as traffic speed (these data are not currently available at a national level for Britain).

During the model selection process various candidate models were run with different spatial scales and pseudo-absence (roadkill and random) points and each provided slightly different variable importance and outcomes (Supplemental Informations 5–18). Grilo et al. (2016) also found the relative importance of variables in the model outputs were not materially altered by changes in sample units and spatial scales, whereas the use of different types of background data did affect the importance of certain variables. The national-level HSI used for the road models, for example had a high permutation importance (>50%) when using random pseudo-absence data and a low importance (<10%) when using roadkill pseudo-absence data. The random pseudo-absence data were spread throughout the width of the road buffer and were consequently further from roads than the roadkill pseudo-absence data. This could have adjusted the effect of distance from roads on hedgehog roadkill and subsequently help better target the effect of specific roads. Future monitoring programmes can be used to ground truth the model results by supplying an independent training dataset against which the model predictions and independent observations can be compared.

The use of different types of background data helps to avoid potential recorder bias generating erroneous conclusions. For example it is uncertain whether roadkills were most recorded in areas with a high probability of hedgehog roadkill occurrence, or whether certain individuals or local citizen science projects, gathering numerous records at a local scale generate ‘clusters’ of records (Isaac & Pocock, 2015). Therefore, the use of random pseudo-absence data was expected to be somewhat biased towards heavily sampled areas, while the roadkill pseudo-absence data would spread predictions to unsampled areas where conditions are similar to sampled areas (Phillips et al., 2009). The low correlation found between the density of roadkill records and roadkill predictions indicates that predictions were not exclusive to highly sampled areas and, therefore, accounted for recorder effort.

Conclusions

This work has shown how the collection of roadkill data can provide large datasets that can be used to understand the seasonal variations, environmental determinants, and spatial distribution of hedgehog roadkill. Our findings suggest that drivers should be most vigilant for hedgehogs in July on roads surrounded by a mix of urban and grassland cover. Areas with high probability of roadkill occurrence and persistent reports of roadkill were identified across the entire road network at a spatial resolution fine enough to contribute to local planning and mitigation. Using this information to target roadkill mitigation efforts should improve our ability to alleviate the impact of projected increases in road cover, traffic levels and urbanisation. This is vital if we are to conserve the size and viability of already declining local and national hedgehog populations in Britain.

Supplemental Information

Supplemental Information 1 Raw data and code.

Click here for additional data file.

Supplemental Information 2 Supplementary Material.

Click here for additional data file.

We are extremely grateful to the teams involved in the setup of Big Hedgehog Map, Mammal Mapper, Mammal Tracker, Mammals on Roads, Project Splatter and Suffolk Wildlife Trust and the citizen scientists involved in the collection of hedgehog roadkill records over the years. We would also like to thank Emily Wilson and Emily Marnham for collating the roadkill data for this project.

Additional Information and Declarations

Competing Interests

Author Contributions

Data Availability

Patrick Graham Ross Wright is employed by the Mammal Society. Frazer Guy Coomber and Prof Fiona Mathews are both honorary members of the Mammal Society. Chloe Bellamy is employed by Forest Research.

Patrick G.R. Wright conceived and designed the experiments, analysed the data, contributed reagents/materials/analysis tools, prepared figures and/or tables, authored or reviewed drafts of the paper, approved the final draft.

Frazer G. Coomber conceived and designed the experiments, contributed reagents/materials/analysis tools, authored or reviewed drafts of the paper, approved the final draft.

Chloe C. Bellamy conceived and designed the experiments, contributed reagents/materials/analysis tools, authored or reviewed drafts of the paper, approved the final draft.

Sarah E. Perkins conceived and designed the experiments, authored or reviewed drafts of the paper, approved the final draft.

Fiona Mathews conceived and designed the experiments, authored or reviewed drafts of the paper, approved the final draft.

The following information was supplied regarding data availability:

All data and code are available at Dryad: Wright, Patrick et al. (2019), Predicting hedgehog mortality risks on British roads using habitat suitability modelling, Dryad, Dataset. DOI 10.5061/dryad.ksn02v70h.

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
