# Peer review of "Predicting hedgehog mortality risks on British roads using habitat suitability modelling"

_PeerJ, doi:10.7717/peerj.8154_

## Round 0.1 · original submission · Major Revisions

Please revise your manuscript, paying close attention to the comments of both reviewers.

Reviewer 1 ·

Basic reporting

- The language is clear, professional and easy to understand.
- The authors give a very good introduction to road ecology and describe the problems associated with roadkills in a very succinct way. However, I would recommend that citizen science be described either in the introduction or later in the material and methods section. The advantages and disadvantages of this method and its influence on the data. Additionally, literature references are okay in general, although more sources on citizen science and how it is used to study roadkills could be added.
- The article is structured as usual. However, I noticed that the authors don't mention all supplementary materials in the text. Thus, references to Supplementary 3, 4, 6 and 7 are missing in the text. Please add this accordingly or reduce the Supplementary materials. Besides, Supplementary 7 is missing completely.
- Figure 1: For me personally this graphic is too overloaded. The variables could be removed and placed in a separate table. Additionally, I would describe the properties of the variables in such a table, because they are missing in the text. For example, I am not familiar with the difference between improved and rough grassland or the characteristics of a "B road". I would like to ask you to add this as it will make the results of the study easier to understand. Especially as a person outside the UK, the road classes are not self-explanatory.
- Figure 2: From my point of view, a classic bar chart would be more appropriate here. For example, average number of hedgehogs per month. But this is optional and probably a subjective assessment.
- The study seems to be self-contained.

Experimental design

- The research presented is within the scope and aims of PeerJ.
- From my point of view, the scientific question is very well chosen and can be answered with the methods presented. The research gap has also been highlighted very well in the introduction.
- In principle, the methods are described appropriately and at a high technical level. Nevertheless, I miss some important details to make the study reproducible.
o The data were collected in various citizen science projects. Please describe briefly what the data collection looks like in the individual projects and how they may differ from each other. If the data collection differs, how were these differences addressed in the analysis?
o How many years of data collection were done, you mention from 2000 onward? Again, this probably differs from project to project, please describe this in the text.
o How do the different projects geographically cover the nation? Are there strong differences? If so, does this have a strong influence on the data?
o In general, the sources of the data should be described in detail. For example, where does the data for variables such as landcover, street properties, etc. come from?
o Line 91: Just a little note, what is NBN gateway?

Validity of the findings

- Raw data: Only the x- and y-coordinates can be found in the Excel table. In order to be able to understand the study, many things would have to be added here. For example, I lack the number of individuals per point, the source of the point, since data from several projects with different survey methods have been used. In addition, please add the data on the variables used, such as the landcover.
- Since your study is based on citizen science data, please elaborate also on the citizens behavior and how this could have influenced your results. You start to address this issue in line 257 and 321, but I would suggest to elaborate on that. For example, it could be that it is easier for citizens to find road-killed hedgehogs in suburban areas than on highways and that’s why there is at the end a high probability of road-killed hedgehogs in suburban areas in your results and not because many hedgehogs were killed in this area.

Additional comments

o I do not understand the sentence in line 248-250. Could you please rephrase the sentence?
o Line 294: There is a typo in “measured”.
o Line 335: I think here is a word missing

Reviewer 2 ·

Basic reporting

The manuscript entitled “Predicting hedgehog mortality risks on British roads using habitat suitability modeling” provided a good overview about the factors related to hedgehog road mortality in Britain and showed some recent approach with habitat suitability models. The importance of this study is that it used a species as a model that suffers a population decline. And, it used a citizen science database, that provided a large amount of information and sometimes is neglected.

Experimental design

One important issue is that authors need to standardize the concepts: road collision, roadkills, road casualties. It is better to use the same concept for the same meaning.
I commented below point-to-point. However, there are some general issues. The introduction is well written and pointed out the main information. The authors just need to improve the goals in the last paragraph and to clarify the main hypotheses (for example, I understand that road types represent different traffic volumes, but it is not clear in the manuscript. In some places, minor roads can have more traffic than major ones. It is important to clarify why you are using this road type classification instead of traffic volume, for example).
Methods need some improvement related especially to the description of the database and clarify the way to calculate some metrics (I mentioned below one by one).

Validity of the findings

The findings are very relevant to the understanding of the relationship between roads and hedgehogs. The conclusions are well stated.

Additional comments

2. Between lines 39 and 48, some references can be replaced by some more current ones. Authors can cite some chapters from the Handbook of Road Ecology (van der ree et al. 2015) instead of van der Ree et al. 2011, for example.
3. Lines 45- 49: The sentence is related to the impact of wildlife-vehicle collisions. Then it is important to clarify mitigation measures for what! I recommend including mitigation measures ”for reducing animal-vehicle collisions”...
4. Line 53: Replace “gloal” by global
5. Lines 54-56: I agree that, in general, researchers and stakeholders are more worried about mitigating large mammal collisions, but authors should make a review about small mammals researches and rewrite the beginning of the sentence. Some studies can be cited here about a formal investigation related to small animals. One example is a book of Roads and Ecological Infrastructure: Concepts and Applications for Small Animals (by Kimberly M. Andrews, Priya Nanjappa, and Seth P. D. Riley. 2015). Furthermore, I made a quick search on google scholar and there are a lot of studies considering roads and small mammals.

6. Lines 66-68: There is a lack of reference in this sentence. I suggest to include some example of the use of these models for the general use (first sentence) and the roadkill prediction (last sentence). Some examples for the roadkill context are: Fabrizio et al. 2018 (https://doi.org/10.1007/s10344-018-1241-7) and Santos et al. 2013 (https://doi.org/10.1371/journal.pone.0079967)


7. Lines 84-86: The last paragraph of the introduction could be used to introduce the reader for what the authors did. The goals must be clear and I suggest to improve the paragraph, including some extra information about the study.

8. Line 88: Hedgehog roadkill data: improving the data description. How many records were considered? When were the records collected (which years, for example)? There is “from 2000 onwards”, however, this is not good to understand the duration of data collection.

9. Line 88: Is there any possibility of double counting in the dataset as data came from different projects?

10. Line 95: It is important to understand the question at comment 8 to understand the temporal analysis. Please, clarify this section.

11. Line 121: Sorry, but I do not understand what “B roads” are.

12. Line 130: As the reference is In Review, it is important to explain a little bit more about the method or change the sequence of sentences. Suggestion: firstly, explain how the model works and secondly explain that the approach is developed by Bellamy (In Review).

13. Line 152: How were “collision risk ratio” calculated? Clarify the way that you calculated it and why you did that. Firstly, I understood that it was calculated for all roads, but in the results, it is not clear because different types of roads have different collision risk ratio.

14. Line 168: Please, clarify how this “inter-annual persistence of roadkill” was calculated. Is it just the record presence?

15. Line 181: In Figure 2: Is the y-axis the number of road casualties recorded? Clarify it in the figure.

16. Lines 182-183: What does explain this result: The national, minor roads and all roads models performed well with AUC scores higher than 0.70, while the major roads and B road models performed less well (AUC ≤ 0.6)? It is nothing about what differ these roads.

17. Figure 3 is not cited anywhere, except Figure 3C. If the other parts are not cited, I presume that there are not important for the results. In this figure, there is another question:

First: graphs (Figure3C and Figure3D) do not have a good resolution. Improve them if you will maintain them.
Second: authors are assessing Habitat suitability indices. Then, why are they calling the probability of occurrence in Figure 3 graphs? They did not model occurrence probability and they did not mention it anytime in the manuscript.

18. Lines 213 and 215: It is a correlation level: why did authors indicate R2?
19. Lines 236-238: Authors could include in this part that this information is also important to plan temporal mitigation measures since this animal has a very defined risk season.
20. Line 292: authors can mention that there are studies that tested the effectiveness of mitigation and there are some successful examples.
21. Line 294: Replace “meansured” by measured
22. Line 295: Authors are writing that their models did not correctly predict hedgehog collision risk, as they wrote: “if the model has correctly predicted areas of hedgehog roadkill”. However, they showed the AUC values and the models were good predictors. I suggest rewriting this sentence.
23. Line 307: It depends on the kind of speed data. Some references show speed limit is not a good predictor for road collisions because it does not express the real speed.
24. Line 340: Greater monitoring of road casualties does not mean that mitigation measures will be effective. I recommend rewrite the last sentence.
25. Resolution of figure 5 is not good.
Supplementary information
26. As I pointed out at the beginning of the comments, the information on supplementary 3 is what I would like to read on the Hedgehog roadkill data section.
27. Supplementary 5 has the same problem of “probability of occurrence” concept mentioned in comment 17.
28. Supplementary 20: include the scale.

---

## Round 0.2 · Minor Revisions

Please make minor revisions according to the reviewer's comments.

Reviewer 1 ·

Basic reporting

First of all, I would like to thank the authors very much for responding to all the comments and for writing a very detailed reply letter. In my view, only two of my previous stated comments were not addressed in the revision:

o The data were collected in various citizen science projects. Please describe briefly what the data collection looks like in the individual projects and how they may differ from each other. If the data collection differs, how were these differences addressed in the analysis?

o How do the different projects geographically cover the nation? Are there strong differences? If so, does this have a strong influence on the data?

Experimental design

no comment

Validity of the findings

no comment

Additional comments

- Line 82-85: Please rephrase the sentence. In the current structure, the listing is difficult to understand.
- Line 90: Please change “Heigle” to “Heigl” in both references.
- Line 99: In my view, “ecological resource” is not the appropriate term. I suggest to use “data resource”.
- Line 125: Cannot find the mentioned information about the data sources used in the supplementary. Please add the information as mentioned in the rebuttal letter.
- Line 336-338: How is this information linked to roadkilled hedgehogs?
- Line 351: Please change “supplementary 19” to “supplementary 18”.

---

## Round 0.3 · accepted · Accept

Thanks for your careful attention to the required revisions.